# Implementation of a lung cancer screening initiative in HIV-infected subjects

Jorge Díaz-Álvarez[1], Patricia Roiz[2,3], Luis Gorospe[4], Ana Ayala[4], Sergio Pérez-Pinto[5], Javier Martínez-Sanz[1], Matilde Sánchez-Conde[1], José L. Casado [1], María J. Pérez-Elías[1], Ana Moreno[1], Raquel Ron[1], María J. Vivancos[1], Pilar Vizcarra[1], Santiago Moreno[1‡], Sergio Serrano-Villar[1‡*]

1 Department of Infectious Diseases, Hospital Universitario Ramón y Cajal, Facultad de Medicina, Universidad de Alcalá, IRYCIS, Madrid, Spain, 2 Department of Internal Medicine, Hospital Universitario Ramón y Cajal, Madrid, Spain, 3 Department of Radiology, Hospital Universitario Ramón y Cajal, Madrid, Spain, 4 Department of Internal Medicine, Hospital de Ávila, Ávila, Spain, 5 Department of Infectious Diseases, Hospital Universitario Ramón y Cajal, Madrid, Spain

‡ Co-senior authors.
* sergio.serrano@salud.madrid.org

**Data Availability Statement:** All relevant data are within the manuscript and its Supporting information files.

## Abstract

In this pilot program of low-dose computed tomography (LDCT) for the screening of lung cancer (LC) in a targeted population of people with HIV (PWH), its prevalence was 3.6%; the number needed to screen in order to detect one case of lung cancer was 28, clearly out-weighing the risks associated with lung cancer screening. While data from additional cohorts with longitudinal measurements are needed, PWH are a target population for lung cancer screening with LDCT.

## Introduction

In a number of countries, life expectancy of people with HIV (PWH) has approached that of the general population, as screening, prevention, and treatment of non-AIDS comorbidities can become their main health issue [1, 2]. Among PWH, lung cancer (LC) is a leading neoplasia [1, 3, 4], and its incidence is expected to increase in the next decade [5].

The United States Preventive Services Task Force (USPSTF) first recommended screening for smokers and former smokers, aged 55–80 years, with at least 30 pack-years history, and no more than 15 years after quitting [6]. Recently, the USPSTF enforced more strict criteria, with a wider age interval from 50 to 80 years, and 20 pack-years history [7]. Given the higher burden incidence of LC among PWH and its poorer prognosis [8], as well as the burden of lung comorbidities, there is now a clear debate on optimal LC screening strategies in PWH. Given that a higher prevalence of false-positive findings, compared to HIV-uninfected individuals, is expected in PWH, it has been argued that LC screening could result in more harm than benefits in this population. Here, we report the experience of a LC screening program for PWH, with adapted criteria seeking higher sensitivity.

## Methods

PWH on follow-up in a tertiary hospital between January 2015 and September 2019 were offered LC screening with LDCT. Inclusion criteria were age 45 years or older, 25 pack-year

**Funding:** This work was supported by the Instituto de Salud Carlos III (projects AC17/00019, PI18/00154, COV20/00349, and ICI20/0058), co-funded by European Regional Development Fund "A way to make Europe"; and by a mobility grant and by the Fundación SEIMC-GESIDA (Ayuda SEIMC). The funders had no role in the study design, data analysis, or interpretation of the results.

**Competing interests:** The authors have declared that no competing interests exist.

history of smoking or more, current smokers, or those who quit within 15 years of screening, and absence of a previous LC diagnosis. We registered the following radiological data: presence of lung nodules, enlarged (>1 cm) lymph nodes, coronary calcium score, aortic dilatation, bone marrow attenuation (at the level of vertebral L1 body), lung emphysema, and non-nodular lung opacities. A single low-dose unenhanced CT (60 mA, 120 kV) was performed on every patient without intravenous contrast; CT machine: high-speed, 16-slice CT machine – Philips (Best, The Netherlands) was used without intravenous or oral contrast. The test was considered positive if a noncalcified nodule was more than 500 mm$^3$ and was considered indeterminate if the solid nodule was 50–500 mm$^3$ or if the diameter of non-solid nodule was greater than 8mm. In those subjects with intermediate results, a follow up scan was performed 3 months after first CT. If at that time the lesion had volume doubling time of less than 400 days, the final result was declared to be positive and if not it was considered negative [9]. Our study takes no consideration on second-round scan (although recruitment is still ongoing). In those subjects in which a LC was suspected, a variety of different diagnostic procedures were performed in accordance to each specific situation. Reading and report of the images were conducted by 2 radiologists specialized on chest radiology. Discrepancies in interpretation between the two thoracic radiologists were resolved by consensus. The images were transferred to the workstation where multiplanar reformatted images were obtained. All images were displayed with two different windows for interpretation (lung window '1500 width/600 level' and mediastinal window '400/40'). The Computed tomography lung analysis was performed using 3D synapse software (Fujifilm Medical, Tokyo, Japan).

We analyzed the findings obtained by the first LDCT of each patient. Descriptive analysis was performed using frequency distributions. We used logistic regression to assess the relationship between baseline variables of interest and the diagnosis of lung cancer. Due to the small number of events, the model has not been adjusted for any confounder. All probabilities were two-tailed, and p value of < 0, 05 was considered to indicate statistical significance. Statistical analyses were performed using Stata v. 16.0 (StataCorp LP College Station, TX, USA).

The Ethics Committee (ceic.hrc@salud.madrid.org) approved the study. The study conformed to the principles of the Declaration of Helsinki and the Good Clinical Practice Guidelines and was approved by the local Ethics Committee; study participants gave their written informed consent to participate in the study.

## Results

A total of 141 patients underwent LDCT, of whom 86% were men and 14% were women. Median age was 57 years (25th-75th percentile, 53–60), 87 (62%) with positive HCV antibodies: median nadir CD4 count was 179 cells/uL (75–305), current CD4 count was 666 cells /uL (403–911), and HIV RNA count < 20 copies/mL was seen in 138 (97.1%) subjects. Median pack-year was 34 (25–41), 122 (82%) were active smokers. Radiological abnormalities were common: pulmonary emphysema in 90 patients (64%), lung non-nodular opacities in 29 (21%), lymph nodes > 1 cm in 10 (7%), aortic atherosclerosis in 48 (34%), aortic dilation in 4 (2.8%), and radiological bone marrow attenuation in 21 (15%) (Table 1).

Lung nodules were found in 52 subjects (37%); < 4 mm in 21 (15%), 4–8 mm in 18 (13%) and > 8 mm in 13 (9%). Only 6 nodules were found to be suspicious for cancer, with patients undergoing invasive procedures. In patient number 2, bronchoscopy was performed without obtaining a representative sample. Due to highly malignant suspicion, video-assisted thoracic surgery (VATS) with inferior right lobectomy and right paraesophageal and paratraqueal limphadenectomy was performed with the final diagnosis. In patient number 3, bronchoscopy without diagnosis was firstly performed. A CT guided fine needle puncture aspiration (PAAF)

**Table 1. Patient baseline characteristics.**

| Caracteristics | Values |
|---|---|
| AGE (Median) (IQR) | 57 (54–60) |
| 40–44 (number) (%) | 1 (1%) |
| 45–49 | 9 (6%) |
| 50–54 | 40 (28%) |
| 55–59 | 52 (36%) |
| >60 | 39 (27%) |
| GENDER (Number) (%) | |
| Male | 122 (85%) |
| Female | 20 (14%) |
| TOBACO USE (Number) (%) | |
| Current | 122 (86%) |
| Previous | 12 (9%) |
| Never | 0 (0%) |
| Unknown | 7 (5%) |
| PACK-YEARS (Median) (IQR) | 34 (33–40) |
| HCV (Number) (%) | |
| Active | 12 (9%) |
| Cured | 70 (50%) |
| Clearance | 5 (3%) |
| No infection | 54 (38%) |
| Unknown | 0 (0%) |
| NADIR CD4+ (Cells/μl) (Median) (IQR) | 179 (75–305) |
| CURRENT CD4+ (Cells/μl) (Median) (IQR) | 666 (403–911) |
| HIV Viral load (log) (%) | |
| < 1,57 log | 138 (97,00%) |
| > 1,57 log | 3 (3,00%) |
| LUNG CANCER PATIENTS (n) | 5 (3.5%) |

showing cytologic examination suggestive of malignancy that was followed by a surgical lobectomy which proved the definitive diagnosis. Following diagnostic procedures, a total of 5 cases of LC was made, yielding a prevalence of 3.6% (95% Confidence Interval [CI] 1.5 to 8.3%) and accounting for the number needed to screen to detect one lung cancer as 28 (95% CI 12–66) (Table 2). Histological examination revealed 4 cases of squamous cell carcinoma and 1 adenocarcinoma. Compared to the rest of our cohort, patients with lung cancer had a similar age (both with a median age of 57 years, p = 0.705), a lower median CD4 nadir count (71 [95% CI 43–105] vs. 179 [95% CI 80–309] cells/uL), lower current CD4 count (352 [95% CI 242–517] vs. 672 [95% CI 430–921] cells/uL), and a higher median pack-year (71 [95% CI 50–91] vs. 32 [95% CI 35–40]).

Excluding patients diagnosed with LC, only 4 patients with lung nodules underwent diagnostic procedures. Flexible bronchoscopy was performed in all of them, but only 2 biopsies were taken (1 CT guided biopsy and 1 surgical biopsy). The only related adverse event was due to surgical biopsy in 1 of the patients who suffered prolonged and mild thoracic pain after the procedure.

Among the 48 patients with radiologic evidence of aortic or coronary atherosclerosis, 5 had already known ischemic cardiomyopathy. Excluding these 4 instances, 15 consultations to cardiology department were performed, resulting in 11 ergometries, 2 dobutamine

**Table 2. General characteristics of patients with lung cancer.**

|  | Age | Gender | Smoking habit | Pack-years | Previous IV drug user | HCV status | HIV Viral load (log copies/ml) | Nadir CD4 + (cells/ml) | Current CD4 + (cells/ml) | Cancer Stage at diagnosis | Lung cancer histology |
|---|---|---|---|---|---|---|---|---|---|---|---|
| Patient 1 | 57 | male | Active | 40 | Yes | Clearance | < 1.57 | 43 | 243 | Stage IV (pT1bN3M1c) | Squamous cell carcinoma |
| Patient 2 | 61 | male | Active | 100 | Yes | Cured | < 1.57 | 10 | 352 | Stage IA3 (pT1c N0 M0) | Squamous cell carcinoma |
| Patient 3 | 57 | male | Active | > 40 | Yes | Cured | < 1.57 | 105 | 182 | Stage IA (pT1 N0 Mx) | Squamous cell carcinoma |
| Patient 4 | 59 | male | Active | 60 | Yes | Cured | < 1.57 | 71 | 517 | Stage IIIA (pT2N2M0) | Squamous cell carcinoma |
| Patient 5 | 56 | male | Active | 82 | Yes | Cured | < 1.57 | 180 | 850 | Stage IV (cT4N3M1b) | Adenocarcinoma |

echocardiograms and 1 coronary angiography. Only 1 diagnosis of ischemic heart disease was made by percutaneous coronary intervention with stenting of medial and distal right coronary artery.

## Discussion

In this pilot screening program of LDCT, in a single-centre cohort of PWH, a total of 5 lung cancers were detected among 141, yielding a prevalence of 3.6% (95% CI 1.5–8.3). Overall, subjects with LC had a low CD4 cell count nadir, incomplete immune recovery, and previous HCV infection, probably indicating underlying chronic immunodeficiency and inflammation, and contributing to LC pathogenesis [1, 3, 4, 10, 11] (Table 2).

As stated above, number needed to screen to detect one lung cancer was 28 (95% CI 12–66). Our results show a higher proportion of LC diagnosis than that appreciated in other LDCT screening trials. In the Nederlands-Leuvens Longkanker Screenings Onderzoek (NELSON) trial, 2.6% of participants [12] were diagnosed with LC. In the study by Makinson et al., a prevalence of 2.0% of LC was found [1], and in another cohort study by Brock et al., [13] there was only one LC amongst 678 patients. A possible explanation driving this lower rates of LC in previous study is the younger median age of the participants in the study by Brock et al. (48 years) and Makinson et al. (median age of 49.8 years). However, in NELSON trial, the baseline characteristics were comparable to our cohort, apart from the HIV condition of all our study participants. Lifestyle factors associated with our study population, such as higher prevalence of previous injection drug than in the general population, could explain in part the high prevalence of LC in our cohort. However, the small sample, and the limited number of cancers must be considered when interpreting the results. Also, our study is not powered enough to estimate the number needed to screen to prevent one death from lung cancer, which is a limitation of our study.

PWH tend to have higher rates of abnormal findings on CT. Although this could pose a problem due to increased false-positive results and related negative consequences due to unnecessary invasive procedures, these complications are infrequent [3, 8, 14–16]. Nevertheless, this issue could be resolved with emergent technologies such as volume-based nodule-management protocols, 64-multidetector CT systems and positron emission tomography [15, 17]. Eventually, this novel technology might imply a reduced number of LDCT required for follow-up in these patients, which remains an unanswered question in this field. In our study, we identified no harm with this LC screening program, beyond uncertain long-term consequences of low radiation exposure, which has been estimated as one death of a radiation-related malignancy in 2,500 patients, from what is received during screening [18].

The histological findings deserve some consideration. Our cohort showed a higher burden of squamous cell carcinoma than adenocarcinoma, in striking contrast with that observed in other series of HIV and non-HIV populations [2, 16, 19, 20]. Although this finding could be affected by the low sample in the study, clinical determinants such as advanced immune suppression at diagnosis, the history of intravenous drug use, or the high prevalence of HCV coinfection, may have influenced the outcomes. These factors could also explain the higher prevalence of LC in our cohort, compared to previous studies [1, 3]. One might note that the potential benefit of LDCT is not only limited to the early detection of LC [14, 21]. As illustrated in the description of the LDCT findings, this program resulted in the detection of conditions in which early management could improve outcomes, such as lung emphysema, osteopenia/osteoporosis and coronary calcifications, all of which may benefit from early detection.

Smoking cessation plans are likely the key intervention to reduce the incidence of lung cancer in PWH. It is important to highlight the high cumulative exposure to tobacco in patients with lung cancer [22], as it can be noted in our cohort the high number of pack-years. Of note, all participants diagnosed with lung cancer were active smokers. Also, in this pilot program, LDCT screening of LC in a targeted population of PWH seems to show results that suggests greater rates of diagnosed lung cancer than in the general population [23], which can be explained by the number of comorbidities in PWH as well as the higher pack-years history compared with people without HIV. In the light of our findings and given the greater risk of LC in PWH, we think that LDCT screening could particularly outweigh the risks in this susceptible population.

In conclusion, while data from larger cohorts with longitudinal measurements are needed, our study reinforces the idea that PWH are a target population for lung cancer screening with LDCT.

## Supporting information

**S1 Data.**
(XLS)

## Acknowledgments

We thank all the patients involved, who made this scientific research possible.

## Author Contributions

**Conceptualization:** Luis Gorospe, Ana Ayala, Sergio Serrano-Villar.

**Data curation:** Jorge Díaz-Álvarez, Patricia Roiz, Sergio Serrano-Villar.

**Formal analysis:** Jorge Díaz-Álvarez, Patricia Roiz, Javier Martínez-Sanz, Sergio Serrano-Villar.

**Investigation:** Jorge Díaz-Álvarez, Patricia Roiz, Luis Gorospe, Ana Ayala, Sergio Pérez-Pinto, Javier Martínez-Sanz, Matilde Sánchez-Conde, José L. Casado, María J. Pérez-Elías, Ana Moreno, Raquel Ron, María J. Vivancos, Pilar Vizcarra, Santiago Moreno, Sergio Serrano-Villar.

**Methodology:** Jorge Díaz-Álvarez, Patricia Roiz, Luis Gorospe, Ana Ayala, Sergio Pérez-Pinto, Javier Martínez-Sanz, Matilde Sánchez-Conde, José L. Casado, María J. Pérez-Elías, Ana Moreno, Raquel Ron, María J. Vivancos, Pilar Vizcarra, Santiago Moreno, Sergio Serrano-Villar.

**Project administration:** Sergio Serrano-Villar.

**Resources:** Sergio Serrano-Villar.

**Supervision:** Santiago Moreno, Sergio Serrano-Villar.

**Validation:** Sergio Serrano-Villar.

**Writing – original draft:** Jorge Díaz-Álvarez, Patricia Roiz.

**Writing – review & editing:** Luis Gorospe, Ana Ayala, Sergio Pérez-Pinto, Javier Martínez-Sanz, Matilde Sánchez-Conde, José L. Casado, María J. Pérez-Elías, Ana Moreno, Raquel Ron, María J. Vivancos, Pilar Vizcarra, Santiago Moreno, Sergio Serrano-Villar.

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
