## [Decision Letter · Decision Letter 0]

30 Mar 2021

PONE-D-21-03298

Implementation of a lung cancer screening initiative in HIV-infected subjects

PLOS ONE

Dear Dr. Serrano-Villar,

Thank you for submitting your manuscript to PLOS ONE. After careful consideration, we feel that it has merit but does not fully meet PLOS ONE’s publication criteria as it currently stands. Therefore, we invite you to submit a revised version of the manuscript that addresses the points raised during the review process.

We look forward to receiving your revised manuscript.

Kind regards,

Michael Cummings, PhD

Academic Editor

PLOS ONE

Journal Requirements:

"Aproved by the " Comite de Ética de la Investigación" del Hospital Universitario Ramón y Cajal (ceic.hrc@salud.madrid.org) ".  

a. Please provide additional details regarding participant consent.

In the ethics statement in the Methods and online submission information, please ensure that you have specified (i) whether consent was informed and (ii) what type you obtained (for instance, written or verbal, and if verbal, how it was documented and witnessed).

If your study included minors, state whether you obtained consent from parents or guardians. If the need for consent was waived by the ethics committee, please include this information.

Reviewers' comments:

Reviewer's Responses to Questions

**Comments to the Author**

1. Is the manuscript technically sound, and do the data support the conclusions?

Reviewer #1: Partly

Reviewer #2: Partly

2. Has the statistical analysis been performed appropriately and rigorously? 

Reviewer #1: I Don't Know

Reviewer #2: I Don't Know

3. Have the authors made all data underlying the findings in their manuscript fully available?

Reviewer #1: Yes

Reviewer #2: Yes

4. Is the manuscript presented in an intelligible fashion and written in standard English?

Reviewer #1: Yes

Reviewer #2: Yes

5. Review Comments to the Author

Reviewer #1: This manuscript is a pilot evaluation of LDCT using adapted screening criteria for people with HIV. Given the high rates of smoking among PWH and the increased lung cancer-specific morbidity and mortality in this population, descriptions of screening outcomes in this population (especially at younger ages than USPSTF guidelines) are needed.

Introduction:

-The USPSTF recently published updated guidelines for LCS in JAMA. The criteria are now 50-80 years of age and minimum 20 pack year history. Obviously, this study was conducted prior to the publication of these new criteria. But in paragraph 2 of the Intro, you should specify that these are the original screening criteria.

Results:

-Line 89, what kind of invasive procedures? Would be helpful to list at least one example of subsequent procedures for these 6 patients.

-On line 90, the authors report that the "number needed to screen" NNS is 28. However, this needs to be specifically defined as "the number needed to screen to detect one lung cancer." NNS is typically reported (i.e., as in the NLST) as the "number needed to prevent one death" which is not what you are referring to here. So please define NNS.

-For Table 1, please provide more detail than "Values" for the top of the table. One column is not labeled.

-In lines 92-95 you are providing outcomes comparing the cancer group to the rest of the cohort. Were these comparisons conducted statistically? Please report stats.

-This is not a suggested revision, but just a comment, that I am overwhelmed at the pack year history reported in the lung cancer patients and their relatively young ages. Screening and tobacco treatment efforts in this population are desperately needed.

Discussion:

-Again, on line 105, the authors report their NNS as 28. This needs to be specified as the "number needed to screen to detect one cancer." Further, comparing 28 in this study to 320 in the NLST is an inappropriate comparison. In the NLST study, NNS was defined as "number needed to screen to prevent one death from lung cancer" and was calculated as the reciprocal of the reduction in the absolute risk of death from lung cancer in one group as compared with the other. You could compare prevalence rates between your study and NLST. But not NNS. Are you able to calculate "number needed to screen to prevent one death from lung cancer" from your dataset? If so, this should be calculated and reported. If not, please list this as a limitation.

-I appreciate the line re: smoking cessation being the key intervention to reduce incidence. Could you expand on this a little more? Perhaps tie it back in to the pack year history observed in your lung cancer group and the fact that all cancer patients were current smokers. You could also cite Tanner et al. 2016 (DOI: 10.1164/rccm.201507-1420OC)

-On Line 141 it is stated "LDCT screening of lung cancer in a targeted population of PWH resulted in greater rates of lung cancer than in the general population." I think this statement needs to be clarified and I'm not sure it is totally accurate. What is the comparison here? The NLST? If so, this is an inappropriate comparison as described above. If you are referring to the general population incidence, then please specify the rates in the population. And do you mean "results in greater rates of [diagnosed] lung cancer"?

-As previously mentioned, there are new LCS criteria. Your study still recruited a younger age (45) than the current guidelines for the general population, but your pack year history (25) is actually higher than the new criteria. I think it would be worth discussing your findings in light of these new guidelines.

Reviewer #2: Thank you for letting me review the research paper by Serrano-Villar and colleagues entitled “implementation of a lung cancer screening initiative in HIV-infected patients”. The paper is well written and of value, and implementation of lung cancer screening in specific populations important to publish, as to assess feasibility of strategies.

May I also apologize here for the delay.

I do have some major comments.

- The reading and report of the images were conducted by 2 radiologists specialized on

62 chest radiology. How were discrepancies managed ?

- Screening procedures are not detailed at all. Specifically is this a single low dose chest tomography, or repetitive each year in case of negative results. What procedures were followed for lung cancer work-up.

- 62% with HCV positive serology, thus I suppose a cancer high risk population, as reflected as 82% of active smokers. I suppose HIV risk in this cohort were primarily due to IDU. This may explain some discrepancies between your study and the published studies (Brock et al. JTO), and more so with Makinson et al. AIDS.

- You do not discus you results in perspective of these previous HIV-screening trials, nor with the NELSON results.

- I would be more careful in my conclusion. What arguments do you have for writing that you participants had impaired immune response, as the median levels of CD4 cells was 666. I suppose you assume as these people had low nadir CD4, their immune system is still very impaired, but I do not think your data shows this. Moreover, nearly all subjects had controlled HIV-disease. You could also add the CD4/CD8 ratio to show that some subjects did not fully restore immune function.

- A descriptive synthetic table of the 5 cancer cases could be interesting with stage at diagnosis, age, etc…I wonder if any cases occurred < 50 years. Should PLHIV at risk be screened a such young ages. Stage at diagnosis is essential, because it may be that diagnosing cancer at advance stages may not be at all beneficial.

- Why so few people between 45-49 were included, as median age of most HIV population is around 50? The pack-years necessary was low, and so the threshold does not seem to me to be an explanation.

- Discussion could be enriched on false positive management : how many had additional procedures, and specifically surgery. How many underwent biopsies…and had adverse events from procedures ? And even bettr, how many people had additional procedures outside the lung cancer work-up procedures (i.e. cardiologist and coronary artery calcification). These are important outcomes in any lung cancer screening strategies.

- Please be clear on the numbers needed to be screened. I believe there is a mistake in the text. In NLST, NNS to detect one lung cancer is 27, and to prevent one lung cancer death 300, which are two very different measures.

- Any data on adherence on the study protocol. Is screening validated in Spain, as clinical practice, and does not necessitate a clinical trial.

- Please mention low numbers, and though 3.6% is an important proportion of subjects with lung cancer, this number is subject to high variability as total numbers were low…(you could give us an CI95%)

Minor comments : why is the NLST trial referenced in the introduction at the end of the fist sentence, as you underscore there the fact that PLHIV have comorbidities as their main issue. For reference 7, and the higher burden of lung cancer in PLHIV I would suggest looking at studies calculation standardized incidence rate ratios…For instance Engels Aids 2006 ; (2) Dal Maso Br J Cancer 2009 ; (3) Grulich Lancet 2007 (4) Bedimo JAIDS 20095 ; (5) : Van Leuween Aids 2009 (6) Robbins Aids 2014 ; (7) Hleyhel CID 2013 ; (8) Hleyhel Aids 2014 ; (9) Hernandez-Ramirez Lancet HIV 2017; (10) Shiels CID 2017.

6. PLOS authors have the option to publish the peer review history of their article (what does this mean?). If published, this will include your full peer review and any attached files.

Reviewer #1: No

Reviewer #2: No

---

## [Author Response · Author response to Decision Letter 0]

22 Jul 2021

Response to Reviewers

Title: Implementation of a lung cancer screening initiative in HIV-infected subjects 

The authors would like to thank the Reviewers and Editors for their careful review of our manuscript, and for providing us with their very helpful comments and suggestions to improve the quality of the manuscript. The following responses have been prepared to address all of the referees’ comments in a point-by-point fashion.

Journal Requirements:

Authors: we have revised the style and followed PLOS ONE's requirements.

a) Please provide additional details regarding participant consent:

 - Authors: we have included additional information. 

b) Once you have amended this/these statement(s) in the Methods section of the manuscript, please add the same text to the “Ethics Statement” field of the submission form (via “Edit Submission”

- Authors: This statement has been added in the Methods section and in the corresponding field of the submission form. 

1. Is the manuscript technically sound, and do the data support the conclusions?

Reviewer #1: Partly

Reviewer #2: Partly

- Authors: We have made our best to carefully report and analyse the data and provide a fair interpretation of our findings. We think that the Review process has significantly improved this new version of the manuscript. 

2. Has the statistical analysis been performed appropriately and rigorously?

Reviewer #1: I Don't Know

Reviewer #2: I Don't Know

This study is mainly descriptive, so the scope of statistical inference is limited. We have detailed the statistical methods in the discussion. The descriptive analysis was performed using frequency distributions. We used logistic regression to assess the relationship between baseline variables of interest and the diagnosis of lung cancer during follow-up. Due to the small number of events, the model has not been adjusted for any confounder. All probabilities were two-tailed, and p value of < 0, 05 was considered to indicate statistical significance. Statistical analyses were performed using Stata v. 16.0 (StataCorp LP College Station, TX, USA). 

3. Have the authors made all data underlying the findings in their manuscript fully available?

Reviewer #1: Yes

Reviewer #2: Yes

4. Is the manuscript presented in an intelligible fashion and written in standard English?

Reviewer #1: Yes

Reviewer #2: Yes

5. Review Comments to the Author:

Reviewer #1: 

- Introduction:

The USPSTF recently published updated guidelines for LCS in JAMA. The criteria are now 50-80 years of age and minimum 20 pack year history. Obviously, this study was conducted prior to the publication of these new criteria. But in paragraph 2 of the Intro, you should specify that these are the original screening criteria.

- Authors: The point raised by the Reviewer is well taken. Indeed, our study was conducted before the publication of the updated USPSTF guidelines for LCS. We have made clear this point in the introduction and also added the reference of the updated paper in JAMA.

- Results:

Line 89, what kind of invasive procedures? Would be helpful to list at least one example of subsequent procedures for these 6 patients.

- Authors: Thanks for the suggestion. We have followed the suggestion and have detailed the diagnostic procedures performed in two cases.

On line 90, the authors report that the "number needed to screen" NNS is 28. However, this needs to be specifically defined as "the number needed to screen to detect one lung cancer." NNS is typically reported (i.e., as in the NLST) as the "number needed to prevent one death" which is not what you are referring to here. So please define NNS.

- Authors: Thank you for noting this inaccuracy. We have followed this suggestion. 

For Table 1, please provide more detail than "Values" for the top of the table. One column is not labeled.

- Authors: Thank you, we have reviewed the headings at include a new column “Cancer stage at diagnosis”. 

In lines 92-95 you are providing outcomes comparing the cancer group to the rest of the cohort. Were these comparisons conducted statistically? Please report stats.

- Authors: We have used logistic regression to explore the associations between the diagnosis of cancer and the general characteristics. Because the sample size in the cancer group is very small (n=5), to avoid overstating the significance of the p values (all were non-significant), we have listed the confidence intervals and described the results: The new paragraph reads as follows:

“Compared to the rest of our cohort, patients with lung cancer had a similar age (both with a median age of 57 years, p=0.705), a lower median CD4 nadir count (71 [95% CI 43-105] vs. 179 [95% CI 80-309] cells/uL), lower current CD4 count (352 [95% CI 242-517] vs. 672 [95% CI 430-921] cells/uL), and a higher median pack-year (71 [95% CI 50-91] vs. 32 [95% CI 35-40])(Table 1)”.

- Discussion:

Again, on line 105, the authors report their NNS as 28. This needs to be specified as the "number needed to screen to detect one cancer." 

Further, comparing 28 in this study to 320 in the NLST is an inappropriate comparison. In the NLST study, NNS was defined as "number needed to screen to prevent one death from lung cancer" and was calculated as the reciprocal of the reduction in the absolute risk of death from lung cancer in one group as compared with the other. You could compare prevalence rates between your study and NLST. But not NNS. Are you able to calculate "number needed to screen to prevent one death from lung cancer" from your dataset? If so, this should be calculated and reported. If not, please list this as a limitation.

- Authors: Thank you for this thoughtful comment. We have specified that we calculated that the number needed to screen to detect one cancer throughout the manuscript. Unfortunately, our study has not enough statistical power to provide accurate estimates of the number needed to screen to prevent one death from lung cancer. We have acknowledged this limitation in the discussion. 

I appreciate the line re: smoking cessation being the key intervention to reduce incidence. Could you expand on this a little more? Perhaps tie it back in to the pack year history observed in your lung cancer group and the fact that all cancer patients were current smokers. You could also cite Tanner et al. 2016 (DOI: 10.1164/rccm.201507-1420OC)

- Authors: We have outlined the importance of smoking as one of the main drivers in LC and highlighted that within LC patients packs-year is substantially higher than in the rest of the cohort. Thank you for suggesting the citation by Tanner et al., that we have properly referenced.

On Line 141 it is stated "LDCT screening of lung cancer in a targeted population of PWH resulted in greater rates of lung cancer than in the general population." I think this statement needs to be clarified and I'm not sure it is totally accurate. What is the comparison here? The NLST? If so, this is an inappropriate comparison as described above. If you are referring to the general population incidence, then please specify the rates in the population. And do you mean "results in greater rates of [diagnosed] lung cancer"?

- Authors: Following this comment, we have discussed in more detailed our findings in the context of previous studies. A new reference has been added to better frame our results. 

As previously mentioned, there are new LCS criteria. Your study still recruited a younger age (45) than the current guidelines for the general population, but your pack year history (25) is actually higher than the new criteria. I think it would be worth discussing your findings in light of these new guidelines.

- Authors: We intended to be more permissive than the LCS criteria that were then enforced. In the light of our findings and given the greater risk of LC in PWH, we think that LDCT screening could particularly outweigh the risks in this susceptible population. However, the performance of the less restrictive new USPSTF criteria should be addressed in next studies in PWH. To give consideration to this Reviewer’s comment, we have included this ideas in the closing remarks in the discussion. 

Reviewer #2: 

The reading and report of the images were conducted by 2 radiologists specialized on chest radiology. How were discrepancies managed?

- Authors: Discrepancies in interpretation between the two thoracic radiologists were resolved by consensus. We have included this information in the methods. 

Screening procedures are not detailed at all. Specifically is this a single low dose chest tomography, or repetitive each year in case of negative results. What procedures were followed for lung cancer work-up. 

- Authors: In this analysis, only a minority of patients had undergone a follow-up CT scan. So, we analyzed the findings obtained by the first LDCT of each patient. We are planning to perform a subsequent analysis with the follow-up data in the future. Following this comment, we have specified this information in the methods. Also, we have detailed the radiological criteria to consider the LDCT finginds suspicious of malignancy, that were those established in the NELSON trial by Van klaveren et al.

62% with HCV positive serology, thus I suppose a cancer high risk population, as reflected as 82% of active smokers. I suppose HIV risk in this cohort were primarily due to IDU. This may explain some discrepancies between your study and the published studies (Brock et al. JTO), and more so with Makinson et al. AIDS. You do not discus you results in perspective of these previous HIV-screening trials, nor with the NELSON results.

- Authors: We agree with the Reviewer. Not only HIV itself or the linked immune suppression, but also several lifestyle factors such as previous IDU could explain the high prevalence of LC in our cohort. 

As requested by the Reviewer, we have compared our findings in the framework ot previous screening trials and cohort studies, and we have included the above-mentioned consideration as a possible factor affecting the results. 

I would be more careful in my conclusion. What arguments do you have for writing that you participants had impaired immune response, as the median levels of CD4 cells was 666. I suppose you assume as these people had low nadir CD4, their immune system is still very impaired, but I do not think your data shows this. Moreover, nearly all subjects had controlled HIV-disease. You could also add the CD4/CD8 ratio to show that some subjects did not fully restore immune function. 

-Authors: We understand the point raised by the Reviewer’s concern. Our intention was to highlight the characteristics of our 5 cases of LC: very low CD4 nadir, all previous HCV diagnosis, and 3 of them CD4 counts below 500 counts. The sentence was confusing because lead the reader understand that we were referring to the overall population. We have reworded the sentence, to make this point clear. 

A descriptive synthetic table of the 5 cancer cases could be interesting with stage at diagnosis, age, etc…I wonder if any cases occurred < 50 years. Should PLHIV at risk be screened a such young ages. Stage at diagnosis is essential, because it may be that diagnosing cancer at advance stages may not be at all beneficial.

- Authors: We share the concern raised by the Reviewer with regards to the age of cancer onset. This concern actually motivated adapting our screening criteria to include a younger population thatn that recommended by the USPSTF . However, all cases pf lung cancer occurred in subjects with older than 55 years. Following this Reviewer’s suggestion, we have added in Table 2 the CDC stage at diagnosis to complete the description of the 5 cases of LC, where the age is also shown. 

- Patients number 1: Stage IV (pT1bN3M1c)

- Patients number 2: Stage IA3 (pT1c N0 M0)

- Patients number 3: Stage IA (pT1 N0 Mx)

- Patients number 4: Stage IV (pT2N1M1a 

- Patients number 5: Stage IV (cT4N3M1b)

Why so few people between 45-49 were included, as median age of most HIV population is around 50? The pack-years necessary was low, and so the threshold does not seem to me to be an explanation.

- Authors: In our clinics, the younger patients were less motivated to undergo such a screening intervention than the older ones. In addition, in our clinics most of the younger patients are MSM and have a lower prevalence of tabacco use than the older patients, which a larger proportion of IDU as a risk factor for HIV, and a larger prevalence of tobacco use. Also, these older patients with previous IDU and many of them AIDS-related conditions had AIDS in the past and are more willing to participate in this kind of preventive initiatives than younger patients. There may be other socio-demographic factors, and even reasons associated with the managing clinician, which are difficult to analyze in the absence of this information in the dataset. 

Discussion could be enriched on false positive management : how many had additional procedures, and specifically surgery. How many underwent biopsies…and had adverse events from procedures ? And even bettr, how many people had additional procedures outside the lung cancer work-up procedures (i.e. cardiologist and coronary artery calcification). These are important outcomes in any lung cancer screening strategies.

- Authors: The comment raised by the Reviewer is well taken. The risk of false positive results is one of the most critical issues realted to LC Screening programs, even more in PWH who have a higher proportion of radiological stigma. Following this Reviewer’s comment, we have reviewed the clinical records to include this information. 

Excluding patients diagnosed with lung cancer, only 4 patients with lung nodules underwent diagnostic procedures. While flexible bronchoscopy was performed in all of them, only 2 biopsies were carried out (1 TC guided biopsy and 1 surgical biopsy). The only related adverse event related to the surgical biopsy was chronic thoracic pain in 1 of the patients.

Among the 48 patients with radiologic evidence of aortic or coronary atherosclerosis, 5 had already known ischemic cardiomyopathy. Excluding these 4 instances, 15 consultations to cardiology department were performed, resulting in 11 ergometries, 2 dobutamine echocardiograms and 1 coronary angiography. Only 1 diagnosis of ischemic heart disease was made by percutaneous coronary intervention with stenting of medial and distal right coronary artery (Table 1).

Please be clear on the numbers needed to be screened. I believe there is a mistake in the text. In NLST, NNS to detect one lung cancer is 27, and to prevent one lung cancer death 300, which are two very different measures.

- Authors: Thank you for noting this. We have reviewed and amended the related statements. 

Any data on adherence on the study protocol. Is screening validated in Spain, as clinical practice, and does not necessitate a clinical trial.

- Authors: Unfortunately, the screening of lung cancer is not currently implemented in Spain. We have justified at the end of the discussion that the aim was to evaluate the experience of a pilot screening program for PWH with adapted criteria seeking higher sensitivity.

Please mention low numbers, and though 3.6% is an important proportion of subjects with lung cancer, this number is subject to high variability as total numbers were low…(you could give us an CI95%)

Authors: Thank you for this suggestion. We have acknowledged in the Discussion the small number of cases as a limitation, and we have included the 95% confidence intervals for LC prevalente (3.6% [95% CI 1.5 – 8.3]) and for the NNT (28 [95% CI 12-66]).

For reference 7, and the higher burden of lung cancer in PLHIV I would suggest looking at studies calculation standardized incidence rate ratios…

- Authors: We appreciate the suggestion. We have changed the reference nº 7 for Robbins HA, Shiels MS, Pfeiffer RM Egels eA. Epidemiologic contributions to recent cancer trends among HIV-ifected people in the United States. AIDS. 2014; 28 (6): 881-90

---

## [Decision Letter · Decision Letter 1]

21 Sep 2021

PONE-D-21-03298R1Implementation of a lung cancer screening initiative in HIV-infected subjectsPLOS ONE

Dear Dr. Serrano-Villar,

Thank you for submitting your manuscript to PLOS ONE. After careful consideration, we feel that it has merit but does not fully meet PLOS ONE’s publication criteria as it currently stands. Therefore, we invite you to submit a revised version of the manuscript that addresses the points raised during the review process.

We look forward to receiving your revised manuscript.

Kind regards,

Michael Cummings, PhD

Academic Editor

PLOS ONE

Journal Requirements:

Reviewers' comments:

Reviewer's Responses to Questions

**Comments to the Author**

1. If the authors have adequately addressed your comments raised in a previous round of review and you feel that this manuscript is now acceptable for publication, you may indicate that here to bypass the “Comments to the Author” section, enter your conflict of interest statement in the “Confidential to Editor” section, and submit your "Accept" recommendation.

Reviewer #1: All comments have been addressed

2. Is the manuscript technically sound, and do the data support the conclusions?

Reviewer #1: Yes

3. Has the statistical analysis been performed appropriately and rigorously? 

Reviewer #1: Yes

4. Have the authors made all data underlying the findings in their manuscript fully available?

Reviewer #1: Yes

5. Is the manuscript presented in an intelligible fashion and written in standard English?

Reviewer #1: Yes

6. Review Comments to the Author

Reviewer #1: This manuscript is greatly improved. All of my comments have been addressed. Thank you for your thorough revisions. I have a few remaining small edits:

Table 1. The IQR is missing for Pack years.

On line 148, I think a word is missing. I think it should read: “…2.6% of participants were diagnosed with LC.”

On line 189 I think that should read “pack-years” not “packs-year”.

7. PLOS authors have the option to publish the peer review history of their article (what does this mean?). If published, this will include your full peer review and any attached files.

Reviewer #1: No

---

## [Author Response · Author response to Decision Letter 1]

21 Oct 2021

The authors would like to thank the Reviewers and Editors for their careful review of our manuscript, and for providing us with their very helpful comments and suggestions to improve the quality of the manuscript. The following responses have been prepared to address all of the referees’ comments in a point-by-point fashion. 

Journal Requirements:

We have reviewed our reference list and we havent been able to find any anomaly related to the issues highlighted. 

Comments to the Author

1. If the authors have adequately addressed your comments raised in a previous round of review and you feel that this manuscript is now acceptable for publication, you may indicate that here to bypass the “Comments to the Author” section, enter your conflict of interest statement in the “Confidential to Editor” section, and submit your "Accept" recommendation.

Reviewer #1: All comments have been addressed

2. Is the manuscript technically sound, and do the data support the conclusions?

Reviewer #1: Yes

3. Has the statistical analysis been performed appropriately and rigorously?

Reviewer #1: Yes

4. Have the authors made all data underlying the findings in their manuscript fully available?

Reviewer #1: Yes

5. Is the manuscript presented in an intelligible fashion and written in standard English?

Reviewer #1: Yes

6. Review Comments to the Author

Reviewer #1: 

This manuscript is greatly improved. All of my comments have been addressed. Thank you for your thorough revisions. I have a few remaining small edits:

- Table 1. The IQR is missing for Pack years.

We have included IQR in table 1. 

- On line 148, I think a word is missing. I think it should read: “…2.6% of participants were diagnosed with LC.”

We have included the error, the word LC has added to the paper. 

- On line 189 I think that should read “pack-years” not “packs-year”.

We have checked this error and we have changes the term “pack-years” for packs-year”. We have also changed “PACKS-YEAR” in table 1 into “PACK-YEARS”.

We have spotted some minor errors in the paper after a thorough recheck. Below we show these changes we have made as long as you accordant with them. 

- On line 24 and 34, LC has been added in order to explain the meaning of the acronym LC (which in our article means lung cancer)

- On line 42, 44, 46, 123 and 190 the words “lung cancer” has been replaced by the acronym “LC”

- On line 45 and 179 “– “has been changed by a “,”. 

- On line 47, the word “cancer” was missing after lung and before screening so we have added LC to address the issue that screening program is for lung cancer

- On line 155 the word “a” was a mistake and it was meant to be “as” so it has been changed. 

- On line 193, “pack-year” has been changed for “pack-years”.

---

## [Editor Report · Decision Letter 2]

3 Nov 2021

Implementation of a lung cancer screening initiative in HIV-infected subjects

PONE-D-21-03298R2

Dear Dr. Serrano-Villar,

We’re pleased to inform you that your manuscript has been judged scientifically suitable for publication and will be formally accepted for publication once it meets all outstanding technical requirements.

Kind regards,

Michael Cummings, PhD

Academic Editor

PLOS ONE
---

## [Editor Report · Acceptance letter]

15 Nov 2021

PONE-D-21-03298R2 

Implementation of a lung cancer screening initiative in HIV-infected subjects 

Dear Dr. Serrano-Villar:

I'm pleased to inform you that your manuscript has been deemed suitable for publication in PLOS ONE. Congratulations! Your manuscript is now with our production department. 

Kind regards, 

on behalf of

Dr. Michael Cummings 

Academic Editor

PLOS ONE